# Psychological Responses to Home-Working Practices: A Network Analysis of Relationships with Health Behaviour and Wellbeing

**DOI:** 10.3390/bs14111039

**Published:** 2024-11-05

**Authors:** Samuel Keightley, Ayla Pollmann, Benjamin Gardner, Myanna Duncan

**Affiliations:** 1Department of Psychology, Institute of Psychiatry, Psychology and Neuroscience, King’s College London, De Crespigny Park, London SE5 8AF, UK; samuel.1.keightley@kcl.ac.uk (S.K.); ayla.pollmann@kcl.ac.uk (A.P.); myanna.duncan@kcl.ac.uk (M.D.); 2Department of Psychology, University of Surrey, Guildford GU2 7XH, UK

**Keywords:** home-working, health behaviours, psychology, health promotion, wellbeing

## Abstract

Working at home, rather than in the workplace, has been suggested to affect office-based workers’ health and wellbeing. This exploratory, cross-sectional study sought to identify discrete psychological responses to home-working practices and investigate their relationship with engagement in health-related behaviours and wellbeing. A sample of 491 home-workers completed a survey assessing ten psychological responses to home-working (e.g., the ability to ‘switch off’ from work), ten health behaviour indices (e.g., sleep trouble), and seven wellbeing indices. Network Analysis modelled relationships between these variables. Results showed four clusters of psychological responses to home-working practices (representing ‘home-working independence’, ‘home-work transition’, ‘daily work pressure’, and ‘work-day forecasting’). Variables within these clusters linked to health behaviour and wellbeing: perceptions of workload manageability, ability to switch off from work, homeworking autonomy, and planning and organising a home-working day had cascading influences on indicators of health, health behaviours, and wellbeing. The findings point to a complex system of potential health and wellbeing consequences of working from home. Further evidence is needed to establish truly causal relationships; nonetheless, our findings call for the development of public health initiatives and organisational policies to support the adoption of home-working practices to benefit the health and wellbeing of home-workers.

## 1. Introduction

Increased adoption of digital technology in the workplace has facilitated a steady rise in office-based employees ‘working from home’ [1]. The COVID-19 lockdowns of 2020 catalysed home-working [2]. Organisations and employees were required to adapt to remote working, and many found that desk-based work could be conducted as productively at home as in the workplace [3]. Post-pandemic, regular home-working, if only for some of the working week, remains prevalent and has become a new norm [4,5].

While much research attention has been paid to promoting health and wellbeing among office-workers (e.g., [6]), public health initiatives have largely overlooked the potential importance of home-working for health and health promotion (e.g., [7]). Home-working practices can have adverse consequences for individuals’ health-related behaviours [8,9]. These consequences can be understood using the COM-B model, which proposes that all behaviours require capability, opportunity, and motivation [10]. Specifically, while working from home may not affect workers’ motivation or capability to engage in physical activity, it removes important opportunities for physical activity. For example, to access their work tasks, office-based workers must commute to and from the workplace, and commuting incurs physical activity [11]. When working from home, however, the activity opportunity afforded by commuting is removed, so physical activity declines [12]. Features of the home-working environment can also affect health behaviour through diminished opportunities. A qualitative investigation into office-based workers’ experiences during the UK COVID-19 lockdown of 2020, for example, showed that workers reported working in smaller workspaces, which further decreased opportunities for incidental movement [13]. Perhaps consequently, they reported increased sedentary behaviour and decreased daily movement.

Home-working may also impact wellbeing. The Job Demands-Resources model (JD-R) proposes that work-related demands, such as perceived work pressure, can lead to strain, exhaustion, and burnout if a worker lacks the resources to manage such demands [14]. Working at home has the potential to both enhance work demands and diminish coping resources. Keightley et al.’s study showed that, at least during the UK COVID-19 lockdown, home-workers felt heightened pressure to be digitally present during working hours [13]. Commuting has been shown to help workers psychologically demarcate work and home settings [15], so discontinuation of commuting due to home-working can hinder work-life separation. Indeed, home-working is associated with greater work-life conflict [16], work-related rumination, and difficulty switching off [9,17,18]. There may also be links between changes in health behaviour and wellbeing; heightened physical activity has been shown to help people manage anxiety and stress [19], so declines in movement and increased sitting when home-working may hinder workers’ ability to cope with increased perceived work demands. There is, however, little literature directly examining the relationship between psychological responses to home-working, health behaviour, and wellbeing [9,13,17].

Identifying how psychological responses to home-working relate to health behaviour and wellbeing could help to inform the future development of public health initiatives to protect health and wellbeing among employees who work from home. This study sought to quantitatively model relationships between psychological responses to home-working practices, health-related behaviours, and wellbeing. While frameworks exist for understanding the respective determinants of health behaviour and wellbeing among workers [10,14], we constructed our model inductively, based on our previous work regarding home-workers’ reflections on their health behaviour and wellbeing [13]. Specifically, we took as our starting point findings from our qualitative study of experiences of home-working among office-based workers during the COVID-19 pandemic [13] and undertook preliminary work to develop quantitative indicators of key variables that emerged from this analysis. Next, we used these variables to undertake a network analysis of relationships between psychological responses to home-working, health behaviour, and wellbeing. Network Analysis methods were used due to the potential complexity of relationships between the variables under investigation.

Previous research has established that home-working can potentially affect health, health behaviour, and wellbeing [7,9,13]. This study sought to explore how home-working may have such an impact, with a special focus on workers’ psychological responses to home-working. Specifically, this study of UK workers aimed to (a) establish discrete psychological responses to home-working practices, (b) examine how these psychological responses interact with one another, and (c) investigate the potential interconnecting pathways of association between psychological responses of home-working practices and indicators of health behaviour and wellbeing. By so doing, we sought to generate evidence to inform the development of organisational and public health policy to ensure the health and wellbeing of people when working from home.

## 2. Method

All procedures were approved by the King’s College London Research Ethics Committee (references: MRSP-20/21-21628, approval date 20 March 2021; MRSP-20/21-23049, approval date 22 April 2021).

### 2.1. Design, Participants, and Procedure

A cross-sectional survey design was used. UK employees who were working from home at the time of data collection were recruited online via Prolific.com, a participant recruitment platform [20]. A recruitment advert was pasted on Prolific to users whose Prolific profile met the following eligibility criteria: UK resident; fluent in English; Prolific approval rate of at least 98% (i.e., high credibility, as established from participation in previous Prolific surveys); employed full-time; and had not taken part in our preliminary item generation study (described below). The advert contained a link to a study website and questionnaire hosted on Qualtrics, on which participants self-reported meeting an additional eligibility criterion (i.e., working from home every day at the time of survey completion).

Survey data were collected in April 2021, 13 months after the first UK Covid lockdown was announced, during which office-workers were required to work from home [21]. At the time of the present study, some Covid-related restrictions were in place, but workers were able to go to the workplace if necessary [21].

Four-hundred-and-ninety-six participants completed an online questionnaire, for which they received a payment of £1.88. After removing five participants who failed at least two of three attention-check items in the questionnaire, the final sample comprised 491 participants (272 female, 218 male, one non-binary; age range 18–73, M = 36.59, SD = 10.40). Participants were most commonly White (English, Welsh, Scottish, Northern Irish), lived in a household with other people, and worked an average of five days a week. Participants worked in a range of industries, including finance and insurance, government and public administration, and education (see Appendix A for full demographic information).

### 2.2. Measures

All measures were self-reported.

#### 2.2.1. Preliminary Work: Item Generation and Development

We sought to construct quantitative measures of psychological responses to home-working practices with the potential to affect health behaviour or wellbeing, as identified in our earlier qualitative study undertaken during the 2020 UK Covid lockdown [13]. Although the qualitative study identified four thematic clusters, authors SK and BG identified across these four themes a set of 35 core, quantifiable psychological responses (i.e., perceptions, experiences, or behaviours). SK generated a set of self-report items intended to capture the core dimensions of each concept. Proposed items were refined, reworded, and verified iteratively through discussions between SK and BG.

For each of the 35 variables (e.g., ‘Having video on during video calls’), three items were generated, producing 105 total items. Each item was preceded by one of three prefixes (“When working from home…”; “When I finish a home-working day…”; “On days when I work from home…”). The three items per variable respectively captured whether a core behaviour was undertaken (e.g., “…I make sure to have my video on during video calls”; ‘Never’ (1)—‘Always’ (7)), perceived control over that behaviour (“…I am free to decide whether or not to have my video on during video calls”; ‘Strongly disagree’ (1)—‘Strongly agree’ (7)), and perceived pressure to engage in that behaviour (“…My colleagues expect me to have my video on during video calls”; ‘Strongly disagree’ (1)—‘Strongly agree’ (7)).

The 105 items were refined using a separate questionnaire-based dataset collected among 240 home-workers, recruited via the Prolific online recruitment platform, in March–April 2021. The same eligibility criteria used for the current study were used to recruit participants for the preliminary study, which was conducted using an online survey tool (Qualtrics). Exploratory factor analysis, for which sample size sufficiency and sphericity assumptions were met (KMO test 0.80; Bartlett’s χ^2^ = 19767.94, *p* < 0.001 [22,23]), generated a nine-factor structure incorporating 83 items. The remaining 22 items were discarded. We removed an additional item which, in hindsight, we deemed unclear (“When I finish a home-working day, I stay in the space that I was working in”). The remaining 82 items were reduced to 25 items by removing items from each factor iteratively until none could be removed without reliability falling below 0.70 (Table 1). A final factor was created by splitting one four-item factor into two separate concepts (‘Workload manageability’; ‘Ability to switch off’).

#### 2.2.2. Questionnaire Measures

*Psychological Responses to Home-Working Practices*. Table 1 lists items used to assess each of the ten preconceived ‘psychological response’ variables.

*Health-related behaviour.* Self-reported physical activity was measured using moderate physical activity (MPA), vigorous physical activity (VPA), and walking items from the International Physical Activity Questionnaire short form (IPAQ-SF [24]). Standardised physical activity definitions from the IPAQ-SF were presented alongside items for comprehension. Items prompted participants to identify how many days in a working week they engaged in MPA, VPA, and walking (e.g., “Over the past working week, on how many workdays did you do vigorous physical activities?”), and, where appropriate, average hours/minutes spent in these activities per day (e.g., “How much time did you usually spend doing vigorous physical activities on one of those days?”).

The metabolic equivalent of task (MET) was calculated for the three physical activity categories. Pre-specified values (Walking = 3.3; Moderate = 4.0; Vigorous = 8.0) were multiplied by the total minutes in each activity, multiplied by the number of days engaged in this activity in a week (e.g., Walking MET-minutes/week = 3.3 * walking minutes * walking days). A single item developed for this study captured daily movement during the working day (“When working from home, which of the following best describes how physically active you are during the working day?”; “Very inactive” (0)—to “Very active” (7)).

Self-reported sedentary behaviour was measured via the Physical Activity and Sedentary Behaviour Questionnaire (PASB-Q [25,26]). Two items respectively captured sedentary behaviour during work (e.g., “On a typical working day, how many hours do you spend sitting, e.g., whilst completing work-related tasks/activities?”) and leisure time (e.g., “On a typical working day, how many hours do you watch television, use a computer, read, or spend time sitting quietly during your leisure time?”). As per Fowles et al.’s coding instructions [25], eight categorical response options (e.g., “None” (1), “<1 h” (2), “1–2 h” (3) … “Above 6 h” (8)) were converted to a single median value (e.g., 2–3 h = 2.5) to calculate a composite time estimate. Estimated total sedentary behaviour time was calculated by summing work and leisure time sedentary behaviour time estimates. An additional item developed for this study sought to capture the average total sitting time during a home-working day (“When working from home, on average, how many hours are you sat down during a typical working day?”; 0–24 h).

One item from the PASB-Q [25] captured intervals for breaks from sitting (“When sitting for prolonged periods (one hour or more), at what interval would you typically take a break to stand or move around?”). Eight categorical options were provided (e.g., <10 min, 10–20 min, 20–30 min… up to >2 h). Ranged response options were converted to median values (e.g., 10–20 min = 15), and the final response >2 h was coded as 120 [25]. 

Sleep trouble (i.e., poor sleep quality) was assessed using the mean of four items from the Copenhagen Psychosocial Questionnaire (COPSOQ—III [27]) (e.g., “How often have you slept badly and restlessly?”; “Not at all” (1)—“All the time” (5). Responses were coded into scores between 0–100 in increments of 25 (e.g., 0, 25, 50, 75, 100; α = 0.88).

Snacking frequency was captured via three items relating to the frequency of eating between meals in the morning, afternoon, and evening (“On a typical working day, how often did you usually eat something between meals in the [morning/afternoon/evening]?” [28]). General consumption of sweet and savoury snack foods was measured with a fourth item (“Please state how often you consume sweet and savoury snack foods”). Responses to all items were on a five-point scale ranging between ‘Less frequently or never’ (coded as 0), ‘1–3 times per month’ (coded as 24), ‘1–3 times per week’ (coded as 104), ‘4–6 times per week’ (coded as 260), and ‘Daily’ (coded: 360). A total snacking composite score was calculated by summing values and dividing by 52 (α = 0.73).

*Health and wellbeing outcomes.* Self-rated health was measured via a single item from the Copenhagen Psychosocial Questionnaire (COPSOQ–III): “In general, which of the following best describes your health…” (“Poor” (1)—“Excellent” (5) [27]). Responses were converted to scores between 0–100 in increments of 25 (e.g., 0, 25, 50). Stress was measured via the Perceived Stress Scale Short Form (PSS-SF [29]), which comprises four items capturing experienced stress over a one-week period (e.g., “How often have you felt difficulties were piling up so high that you could not overcome them?”; “Never” (1)—“Very often” (5)). One of the four items was adapted to the home-working context: “How often have you found that you could not cope with all the things that you had to do?”. Items were summed to produce a total stress score (α = 0.81).

The COPSOQ-III was used to capture four work-related wellbeing constructs [27]. Cognitive stress (e.g., “How often have you had difficulty in making decisions?”) and Burnout (e.g., “How often have you been emotionally exhausted?”) were measured on four-item scales (“Not at all” (1)—“All the time” (5)). Responses to four work-life conflict items (e.g., “The demands of my work interfere with my private and family life”) ranged from “To a very small extent” (1) to “To a very large extent” (5). We used a single *job satisfaction* item (“Regarding your work in general over the past working week, how pleased are you with your job as a whole, everything taken into consideration?”; “Very unsatisfied” (1)—“Very satisfied” (5). Multi-item-scale scores were averaged to create composite scores (α’s = 0.86–0.91). All four work-related wellbeing variables were converted into 0–100 scores in increments of 25 (e.g., 0, 25, 50, 75, 100).

General wellbeing was measured using the seven-item Short Warwick-Edinburgh Mental Wellbeing Scale [30]; e.g.,: “I’ve been feeling optimistic about the future”; “None of the time” (1)—“All of the time” (5)). Items were summed and converted to a metric score via Stewart-Brown et al.’s conversion table (α = 0.84 [30]).

Perceived isolation was measured using the UCLA Three-Item Loneliness Scale [31] (e.g., “How often do you feel that you lack companionship?”; “Hardly ever” (1); “Some of the time” (2); “Often” (3); α = 0.82).

### 2.3. Data Analysis

Statistical analyses were conducted using R Version 4.2.2 [32] with R Studio version 2022.7.2.576 [33]. We did not expect differences in scores across socio-demographic groups; nonetheless, for completeness, variable scores were compared across socio-demographic characteristics to explore potential differences according to participant gender, industry, ethnicity, and living situation. The results of these analyses are reported in Appendix A.

Harman’s single-factor test, conducted to assess for the presence of common method variance [34,35], indicated that a single factor accounted for 18.61% of the variance (SS loadings = 14.52). This figure was substantially below the 50% threshold, indicating that common method bias was not a significant concern.

Prior to network estimation, to account for non-normal values on four variables (moderate exercise, vigorous exercise, walking, and work-time sedentary behaviour), data was non-paranormal transformed using the npn function in the *huge* R package [36]. Pre- and post-normalisation results are in Appendix A.

To estimate the structure of relationships between variables, exploratory network analyses were undertaken on the ten home-working response variables, ten health behaviour variables, seven wellbeing variables, and two demographic variables (age, job tenure; Appendix A). No data were missing.

#### 2.3.1. Estimating a Network of Relationships Between Responses to Home-Working Practices, Health Behaviour, and Wellbeing

A visual network of relationships between variables based on the Gaussian graphical model was estimated using the R package *bootnet* [37]. Within the network analysis, the variables were visualised as circles. Observed relationships between variables represent partial correlation coefficients between each of the variables in the network [38], estimating the extent of (in)dependence of each variable on neighbouring variables [39]. Relationships in a network were visualised as connecting lines, with positive relationships depicted in blue and negative correlations in red, and thicker lines indicating a stronger relationship [40]. The least absolute shrinkage and selection operator method was used to minimise the risk of spurious associations within the network [37,39]. The extended Bayesian information criterion [41], with a hyperparameter of 0.25, was used for network selection [42]. The chosen network estimation method was applied using the *EBICglasso* function [43,44], and network estimation results were plotted using the *viridis* package [45].

#### 2.3.2. Identifying Clusters of Psychological Responses to Home-Working Practices

Exploratory graph analysis [46], using a weighted network community detection algorithm (*Walktrap*), was used to identify whether distinct clusters existed within the measured responses to working from home [47]. Each extracted cluster was labelled by the authors according to their contributing components.

#### 2.3.3. Identifying Influential Variables and Relationships in the Network

The most important (‘central’) variables and relationships within the estimated network were identified using the ‘expected influence’ centrality index, which reveals the extent to which variables appear to influence one another based on the sum of positive and negative extended connections [48]. Values furthest from zero denote a more important variable, with the direction of association denoted by a positive or negative value. The accuracy of the estimated network and the stability of centrality indices were assessed through non-parametric bootstrapping methods via the *bootnet* package [40], using 2000 randomly allocated bootstrapped samples. Correlational stability coefficients >0.5 were taken to indicate stability [49].

#### 2.3.4. Identifying the Influence of Psychological Responses to Home-Working Practices on Health Behaviour and Wellbeing

The influence of home-working response variables on health behaviour and wellbeing variables was modelled using ‘bridge expected influence’ coefficients, which estimate the potential for that variable to impact multiple variables in the network. Strong ‘bridge expected influence’ values indicate that a variable has greater potential to activate or deactivate a *cluster* of other variables within the network [50,51]. Values furthest from zero demonstrate the potential for the variable to positively or negatively influence other aspects of the network. Shortest pathway diagrams were created to visualise key variables and their direct and indirect associations with other variables in the network [52].

A more detailed, technical account of Network Analysis procedures and results is provided in Appendix A.

## 3. Results

Figure 1A depicts the final estimated network. Network accuracy and stability coefficients were satisfactory, indicating the adequacy of the extracted network.

### 3.1. Identifying Clusters of Psychological Responses to Home-Working Practices

Four clusters of home-working practice responses were observed. Cluster one, which we labelled “home-working independence”, contained three items, relating to perceived home-working autonomy (Variable 2 [V2]), perceived pressure to work at the same time as colleagues (V1), and pressure to attend online meetings (V3). Cluster two (“home-work transition”) comprised three items incorporating the ability to ‘switch off’ from work (V4), perceived control over transitioning between home and working environments (V5), and perceived ability to transform workspaces (V6). Cluster three (“daily work pressure”) comprised two items covering perceptions of daily workload manageability (V7) and perceived excess of daily meetings (V8). Cluster four (“work-day forecast”) combined two items, including perceived pressure to have their video on during meetings (V9) and intentions to plan and organise the work day (V10).

### 3.2. Estimated Relationships Among Psychological Responses to Home-Working Practices and Indicators of Health, Health Behaviour, and Wellbeing

The strongest relationship among individual home-working variables was a negative association (−0.22) between the perceived ability to switch off (V4) and perceived work-life conflict (V11). The home-working response variable that had the strongest relationship with health behaviours or wellbeing was perceived workload manageability (V7), which had a negative association (−0.09) with work-time sedentary behaviour (V26). The strongest relationship observed between any health-related behaviour and wellbeing variable was a positive association (0.17) observed between sleep trouble (V19) and burnout (V14).

*Influential variables within the network.* Of home-working response variables, home-working autonomy (V2; expected influence [EI] −2.67) and perceived daily workload manageability (V7; EI: −2.40) had the greatest influence across the whole network. Influence was markedly lower for the other home-working response variables, though the perceived ability to switch off (V4; EI: −0.93), perceived excess of daily meetings (V8; EI: −0.89), and perceived autonomy over transitioning between home and work environments (V5; EI: 0.84) demonstrated relatively high expected influence. Among wellbeing indicators, general wellbeing (V16; EI: −1.43) and burnout (V14; EI: 1.24) displayed the highest influence. Other wellbeing-related nodes of apparent importance were cognitive stress (V12; EI: 0.59), work-life conflict (V11; EI: −0.55), and job satisfaction (V13; EI: 0.51). Among the indicators of health-related behaviour, similar influence values were found for work-time physical activity (V23; EI: 1.19), vigorous exercise (V20; EI: 0.98), break-taking intervals (V25; EI: −0.88), sleep trouble (V19; EI: 0.83), and moderate exercise (V21; EI: 0.79). All remaining variables demonstrated relatively low influence (EI £½0.5).

*Assessing relationships between clusters in the network.* The three variables that most strongly linked home-working responses with health behaviour or wellbeing clusters were: perceived ability to switch off from work (V4; Bridge Expected Influence [BEI]: −0.42), which was situated in the “home-working transition” cluster; daily workload manageability (V7; BEI: −0.32), which was located in the “daily work pressure” cluster; and work-day planning and organisation (V10; BEI: 0.23), which was in the “work-day forecast” cluster (Figure 1C). Notably, high expected influence values were also identified for work-life conflict (V11; BEI: −0.27), which was in the “wellbeing” cluster, and sleep trouble (V19; BEI: 0.39), which was in the “health and health behaviour” cluster.

Perceived ability to switch off from work (V4) was most associated with the “wellbeing” cluster via its negative relationship with work-life conflict (V11; *r* = −0.22) and was most related to the “health and health behaviour” cluster via a negative relationship with sleep trouble (V19; *r* = −0.08). Daily workload manageability (V7) was most strongly associated with the “wellbeing” cluster via a negative relationship with work-life conflict (V11; *r* = −0.15) and with “health and health behaviour” via a negative relationship with work-time sedentary behaviour (V26; *r* = −0.09). Work-day planning and organisation (V10) was most related to the “wellbeing” cluster via a positive relationship with job satisfaction (V13; *r* = 0.11) and with the “health and health behaviour” cluster via a negative relationship with global sedentary behaviour (V24, *r* = −0.07).

The shortest route of associations was between psychological responses to home-working practices and health behaviour and wellbeing clusters. Variables with the strongest expected influence centrality values from the “wellbeing” and “health and health behaviour” measures were selected to investigate the most direct relationships (i.e., ‘shortest pathways’) with psychological responses to home-working. General wellbeing (V16; EI = −1.43) and burnout (V14; EI = 1.24) were selected from the “wellbeing” cluster, and work-time physical activity (V23; EI: 1.19) and sleep trouble (V19; EI: 0.83) were selected from “health and health behaviour”.

General wellbeing (V16) displayed the shortest pathway of association to psychological responses to home-working practices via a perceived freedom to transition between home and working environments (V5), showing a notable connection with a perceived ability to transform workspaces (V6; Figure 2A). Four other home-working response variables (pressure to work at the same time as colleagues [V1], home-working autonomy [V2], pressure to attend meetings [V3], and work-day planning and organisation [V10]) connected with general wellbeing via an intermediate step, through the job satisfaction variable (V13). The remaining home-working response variables were connected with the strongest wellbeing cluster variables via two intermediate steps. Figure 2B demonstrates the shortest associative pathways from burnout (V14), with only a single direct connection to home-working practice responses via daily workload manageability (V7). This association showed interactions with perceived excess of daily meetings (V8) and video on/off pressure (V9). Burnout (V14) displayed a notable connection with the ability to switch off requiring an intermediate step via work-life conflict (11). The remaining connections between home-working response variables and burnout (V14) were weak or required at least two steps.

Work-time physical activity (V23) required an intermediate associative step between work-time sedentary behaviour (V26) when connecting with perceived daily workload manageability (V7) and the ability to transform workspaces (V6; Figure 2C). Work-time physical activity also displayed an intermediate association with home-working autonomy (V2) and pressure to attend meetings (V3) via break-taking intervals (V25) and displayed an indirect connection to work-day planning and organisation (V10) via self-rated health (V18). The remaining associative pathways to home-working practice responses required two or more intermediate steps. In Figure 2D, the shortest pathway of association between sleep trouble (V19) and most other home-working practice responses was via the ability to switch off (V4). The remaining associative connections between sleep trouble (V19) and home-working practice responses required multiple intermediate steps.

## 4. Discussion

Public health initiatives to promote office-worker health have largely failed to recognise the potential for home-working to detrimentally impact health behaviour and wellbeing. This exploratory study sought to document discrete psychological responses to home-working practices and model their relationships with health-related behaviours and dimensions of wellbeing. We assessed these relationships using items constructed for the purpose of this study rather than standardised measures and adopted a cross-sectional design that precludes assessments of causality. Our results should, therefore, be considered preliminary, and replication is needed. Nonetheless, our findings suggest that the ways in which home-workers psychologically respond to home-work practices may affect their health behaviour and wellbeing.

Using measures of key variables that emerged from our earlier, qualitative research around how workers experience home-working [13], our results indicated that there were four clusters of psychological responses to home-working: perceptions of independence when working at home; the psychological transition between home and work; perceived work pressure; and forecasting the workday. Furthermore, these responses appeared consequential for health and wellbeing. A cluster relating to ‘home-working independence’ encapsulated perceptions of autonomy when working from home, a lack of perceived pressure to work at the same time as colleagues, and a lack of perceived pressure to attend meetings. The observed association between perceived pressure and autonomy echoes previous research that suggests that feeling pressured to be seen to be performing certain work tasks on the same schedule as colleagues can limit perceived autonomy [53,54]. Perceived autonomy at work is a key resource for coping with job demands [14] and, importantly, has previously been linked to workers’ opportunities to move more and sit less during the working day [55], which in turn may boost wellbeing and enhance the perceived ability to cope with job demands (e.g., [19]). We found home-working autonomy to be positively related to work-time physical activity via break-taking intervals, suggesting that workers who felt more independent were more likely to report frequently standing up and moving around during concentrated work periods. Taking breaks has been shown to provide important opportunities for recovery [56], in turn enabling us to better manage workplace demands.

Two clusters of home-working practice responses, relating to feeling able to transition between ‘home’ and ‘work’ mindsets and perceived daily work pressure, had similarly notable relationships with indicators of health behaviour and wellbeing. The “home-work transition” cluster captured workers’ ability to transform their workspace at the end of the home-working day, an ability to ‘switch off’ from work during leisure time, and a perceived freedom to transition between home and working environments. The “daily work pressure” cluster comprised workers’ perceptions that their workload was unmanageable and that excessive online meetings were interfering with their productivity. Within these clusters, the perceived ability to ‘switch off’ was linked to work-life conflict, burnout, and poorer sleep, whilst a manageable workload was associated with a reduction in sedentary behaviour during work hours, less work-life conflict, and reduced burnout. Our findings are important from a theoretical perspective. From the perspective of the Job Demands-Resources model [14], which proposes that workers must possess sufficient workplace and personal resources to cope with job demands, our findings demonstrate a potential interplay between work demands and coping resources. They also show how, in line with the COM-B model of behaviour [10], home-working practices can diminish opportunities for health-promoting behaviours. For example, workload pressure, a key job demand, can increase the time and psychological effort that workers spend on work-related matters in non-work time, reducing psychological recovery from work [16,57]. Furthermore, managing one’s workload involves engaging in more computer-based work, which reduces opportunities for movement and so will likely increase sitting time, and engaging in such health-risk behaviour can compromise the wellbeing needed to cope with heightened work demands [12,19,58]. Moreover, the blurring of work-home boundaries and increases in workload increase the likelihood that workers will ruminate about work during their leisure time [59], which impairs the ability to cope with work stress or to rest and recover [60,61,62]. The combination of increased stress and a lack of rest and recovery increases the risk of burnout [14,63]. In sum, our findings suggest that feeling one’s workload is unmanageable and failing to psychologically detach from work can instigate a cascade of adverse health behaviour and wellbeing outcomes for home-workers.

Surprisingly, the “workday forecast” cluster revealed a positive relationship between feeling able to plan and organise one’s workday and experiencing pressures to have one’s video camera on during work meetings. This relationship was the weakest of all those observed within the four clusters, so it may represent a statistical anomaly rather than a meaningful, unidimensional psychological experience. Nonetheless, some extant evidence highlights the fatigue effects of the camera on digital meetings, indicating a greater associated workload, which possibly requires a heightened need to plan and organise one’s working day [64,65]. We found that intentions to plan the workday predicted greater job satisfaction and general wellbeing. This may reflect that those who intend to plan their day have greater control, agency, and self-efficacy over their working schedule, which have been found to predict both job satisfaction and wellbeing among home-workers (e.g., [5,66]). We also observed a link between feeling able to plan one’s home-working day and reductions in both workday and waking-day sedentary behaviour. This could suggest that people with greater opportunity to plan the day ahead may purposefully limit the time spent on seated tasks, or that those with greater planning and organisation abilities more broadly tend to spend less time sitting [67,68]. While further research is needed to replicate and explain the potential influence of workday planning and organisation on other home-working practice responses and health behaviour, our findings point to the potential for individual differences in psychological responses to home-working practices to influence health behaviour and wellbeing.

Our findings speak to the importance of developing public health initiatives and organisational policies to target home-working practices, or the ways in which people psychologically respond to such practices, to potentially improve the health and wellbeing of home-workers. We urge public health bodies to seek to raise awareness of the potential for home-working to adversely impact health and wellbeing, thereby encouraging employers to adopt more health-conducive home-working policies. Specifically, our findings call for employers to foster among workers a sense of freedom to shape their home-working day, while also endorsing regular break-taking as a means of promoting rest and recovery, physical activity, and productivity, health, and wellbeing. Additionally, employers should seek to adopt policies that minimise non-productive work time (e.g., limiting unnecessary meetings or imposing maximum meeting lengths). Employers should also educate, train, and support their workers to adopt practices that can aid workload management and thus workload demands and encourage them to prioritise detaching from work at the end of the home-working day, facilitating job resources. For example, workers who bookend their working day by engaging in a purposeful activity can experience wellbeing benefits arising from a marked transition from work to leisure time [13]. Where such activities are also directly health-conducive, such as going for a walk (e.g., a ‘fake commute’ [69]), health and wellbeing benefits will likely be heightened [70].

Limitations must be acknowledged. We constructed new questionnaire items for the purpose of this study, based on the reflections of participants in our earlier qualitative study of home-working during the COVID-19 pandemic [13], and reduced these based on new data collected from 240 home-workers. While we derived items for this study systematically using a dataset obtained from an independent sample, the replicability of our findings is not clear. More research is needed to establish standardised measures of key psychological responses to home-working. Furthermore, our study used data collected in April 2021, when UK Covid-19 lockdown restrictions had been eased [21], but the remaining restrictions likely resulted in more office-based workers working at home than before or after the pandemic. Findings may lack generalisability to the more stable, post-pandemic home-working contexts. Additionally, while our statistical analysis estimated directional relationships between variables, the cross-sectional nature of our data precludes conclusions around causality. Similarly, reliance on self-report data is problematic because people typically overestimate engagement in health-promoting behaviours and under-report health risk behaviours [71]. We encourage replications of our study using longitudinal time series methods for more rigorous estimation of home-workers’ psychological responses to working practices and their impact on health behaviours and wellbeing in the post-pandemic context [72].

Our study overlooked the heterogeneity of home-working practices and responses to them. Home-working practices, and their impact on health behaviour and wellbeing, may differ according to job type, role, expectation, and work settings [73,74,75]. For example, unique work cultural factors associated with call centre jobs, such as continuous monitoring of productive and personal time and a high expectation to be digitally present, demonstrate notable impacts on work-time physical activity and prolonged sitting [55]. We also excluded home-workers with caring responsibilities who we deemed likely to have personal circumstances that influence their working practices and, by extension, their health and wellbeing. Yet, one in seven UK workers has caring responsibilities [76]. Future research should seek to better account for the various organisational, situational, and individual characteristics that influence home-working practices and psychological responses to them.

## 5. Conclusions

Our findings move beyond previous research that has shown the impact of home-working on health and wellbeing [7] and demonstrate potential pathways through which home-working may have such an impact. We showed that workers’ psychological responses to home-working practices are linked to health-related behaviours and wellbeing. Specifically, perceptions of workload manageability, ability to switch off from work in the evening, a tendency to plan a home-working day, and home-working autonomy appeared to have cascading effects on health behaviours, such as sitting and physical activity, and aspects of work-related wellbeing such as stress and burnout. More research is needed to replicate the complex causal relationships we observed. Nonetheless, our findings call for the development of public health initiatives to raise awareness of the impact of home-working on health behaviour and organisational policy that supports productive home-working practices conducive to workers’ health and wellbeing.

## Figures and Tables

**Figure 1 behavsci-14-01039-f001:**
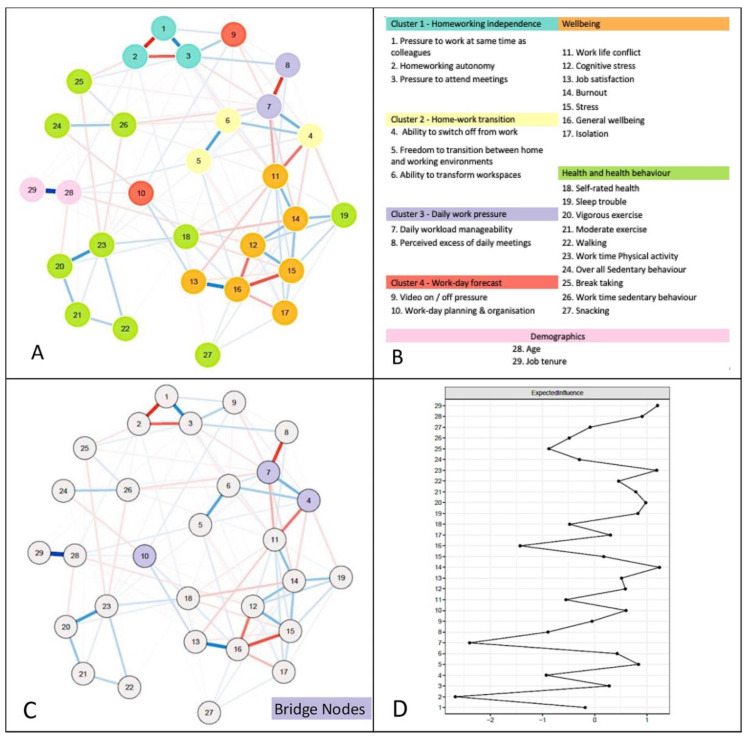
Final estimated network (**A**); final observed clusters and indicators of health, health behaviours, and wellbeing (**B**). variables observed to bridge gaps between clusters (**C**). Centrality indicator expected influence (**D**). Note: (**A**) Overall network including clusters of psychological responses to home-working. Blue lines = positive relationships; red lines = negative relationships. Thicker lines between variables indicate a stronger relationship. (**B**) Representative node names and network cluster names. (**C**) Observed links between psychological home-working practice responses and health behaviour and wellbeing. (**D**) Standardised Z-scores for node expected influence centrality metric. Nodes furthest away from the value of zero indicate a high expected influence.

**Figure 2 behavsci-14-01039-f002:**
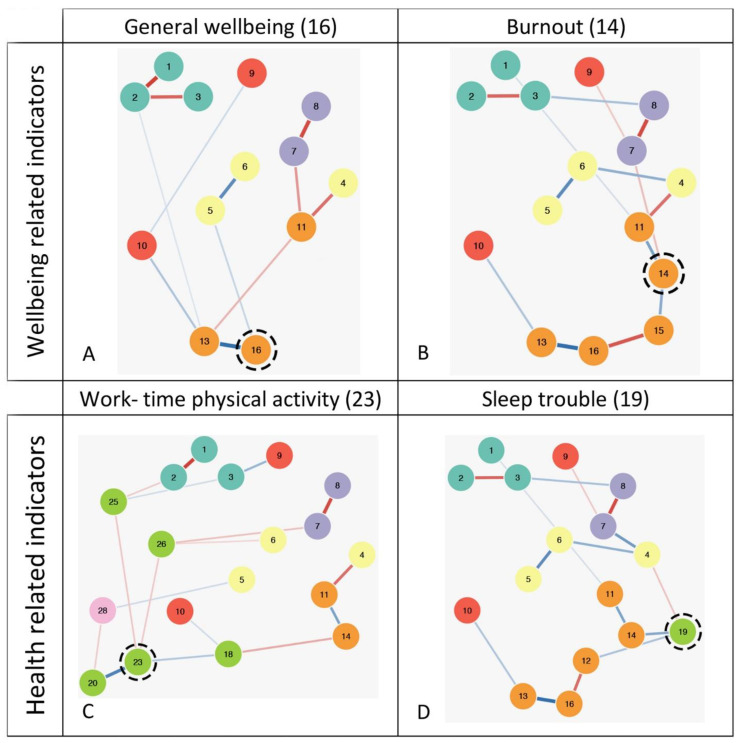
*Isolated networks of direct pathways between the identified most influential wellbeing-related and health-related nodes. Note:* (**A**) Network depicting shortest paths between node 16 (general wellbeing) to home-working responses (nodes 1–10). (**B**) Network depicting shortest paths between node 14 (burnout) to home-working responses (nodes 1–10). (**C**) Network depicting shortest paths between node 23 (work-time physical activity) to home-working responses (nodes 1–10). (**D**) Network depicting shortest paths between node 19 (sleep trouble) to home-working responses (nodes 1–10). Blue lines = positive partial correlations; red lines = negative partial correlations. The thickness of the edge between nodes indicates the size of the correlation (the thicker the line, the stronger the connection). See Figure 1B for representative node names.

**Table 1 behavsci-14-01039-t001:** Psychological responses to home-working practices and representative items.

Psychological Home-Working Responses *(Prefix Number)* *	Item
Pressure to work at same time as colleagues (2)	My colleagues expect me to work the same amount of hours as they do
	I feel I have to work at the same time as my colleagues
	I have no choice but to work for the same amount of hours as my colleagues
Homeworking independence (2)	I am free to choose when I am online during the working day
	I am free to complete my work tasks at whatever time I wish
	I can choose how many hours to work each day
Pressure to attend meetings (2)	My colleagues expect me to attend all work-related meetings
	I feel pressured to attend all work-related meetings
	I am free to decide how many relevant work-related meetings I attend
Ability to switch off (3)	I think about work long after I have finished the working day
	I am able to disconnect from work after I have finished the working day
Freedom to transition between home and working environments (4)	I am able to leave the space that I was working in
	I am able to move to a space that I don’t use for work
Ability to transform workspaces (2)	It is difficult for me to pack away my work-related items (e.g., computer, documents etc.) when I finish the working day
	I am free to pack away my work-related items (e.g., computer, documents etc.) when I finish the working day
	I have no choice but to leave my work-related items (e.g., computer, documents etc.) set up when I finish the working day
Workload manageability (2)	I have enough time in the day to deal with my workload
	It is difficult for me to take breaks
Perceived excess of daily work meetings (2)	I feel that the amount of meetings I have is excessive
	I feel that I attend too many meetings
Video on/off pressure (2)	My colleagues expect me to have my video on during video calls
	I feel that I should have my video on during video calls
	I am free to have my video off during video calls
Work-day planning and organization (3)	I make sure to plan out each working day
	I plan each day to make sure I get all my work done

*Note. ** Prefix statements were as follows: Prefix 1 = ‘When working from home…’ (frequency scale); Prefix 2 = ‘When working from home…’ (agreement scale); Prefix 3 = ‘On days when I work from home…’ (frequency scale); Prefix 4 = ‘When I finish a homeworking day…’ (frequency scale).

## Data Availability

The data presented in this study are available at https://osf.io/xtb3v/ (accessed on 3 October 2024).

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
