# Peer review of "Psychological Responses to Home-Working Practices: A Network Analysis of Relationships with Health Behaviour and Wellbeing"

_behavsci, 2024, doi:10.3390/bs14111039_

Round 1

Reviewer 1 Report

Comments and Suggestions for Authors

Thanks for the opportunity to review the manuscript.

I would like to congratulate the authors for writing this paper. This manuscript is interesting and the choices made in the research seem to be relevant. 

However, for publications, this paper requires some changes/additions/ and edits. I hope these comments and suggestions may be useful for improving your study. Good luck!

- INTRODUCTION: A greater effort is required to identify the contributions derived from this study and the relevance of the study.

- LITERATURE REVIEW: it would be important to have a chapter with references to relevant authors in the area, who justify the study.

- DISCUSSION: The discussion of results is interesting. Would be better if you link the results obtained with the literature review.

- CONCLUSIONS: Reference should be made to the authors referred to in the literature review, indicating whether the conclusions of this study are in accordance with them. Beyond the result discussion, it is necessary to enhance both the theoretical and practical contributions that this article implies to current literature.

Author Response

R1 COMMENT 1:

INTRODUCTION: A greater effort is required to identify the contributions derived from this study and the relevance of the study.

OUR RESPONSE TO R1 COMMENT 1: We now include in the Introduction references to important theory regarding the determinants of health behaviour (the COM-B model; Michie et al, 2011) and worker wellbeing (Bakker & Demerouti, 2017). We also provide a statement at the end of the Introduction that explicit states the intended contribution of the study (lines 93-100):

this study of UK workers aimed to (a) establish discrete psychological responses to home-working practices, (b) examine how these psychological responses interact with one another, and (c) investigate the potential interconnecting pathways of association between psychological responses of home-working practices and indicators of health behaviour and wellbeing. By so doing, we sought to generate evidence to inform the development of organizational and public health policy to ensure the health and wellbeing of people when working from home.”

R1 COMMENT 2:

LITERATURE REVIEW: it would be important to have a chapter with references to relevant authors in the area, who justify the study.

OUR RESPONSE TO R1 COMMENT 2: We note that Behavioral Sciences does not require a standalone ‘literature review’ section of the paper. Instead, in keeping with journal requirements, we have included an Introduction that explains the research gap that the study fills. The Introduction also now includes additional relevant references to theory (e.g., COM-B model; Job Demands-Resources Model), and references regarding home-working (e.g., Oakman et al., 2020). All references are relevant to the rationale for our study.

R1 COMMENT 3:

DISCUSSION: The discussion of results is interesting. Would be better if you link the results obtained with the literature review.

OUR RESPONSE TO R1 COMMENT 3: We now better link our results to the Introduction. Specifically, we interpret the findings in light of the COM-B model and Job Demands-Resources model (lines 462-467). We also include a newly expanded, standalone paragraph on the implications of the findings for policy and practice (lines 504-522), so explicitly addressing the overarching aim of the study, as now stated at the end of the Introduction (lines 93-100).

R1 COMMENT 4:

CONCLUSIONS: Reference should be made to the authors referred to in the literature review, indicating whether the conclusions of this study are in accordance with them.

OUR RESPONSE TO R1 COMMENT 4: We have respectfully chosen not to indicate, in the Conclusion, whether our study findings agree with previous authors. (This is done throughout the Discussion section.) Instead, as is more conventional for a Conclusion section, we focus on summarising our findings and their implications for policy and practice.

R1 COMMENT 5:

Beyond the result discussion, it is necessary to enhance both the theoretical and practical contributions that this article implies to current literature.

OUR RESPONSE TO R1 COMMENT 5: We now relate our study to theory in the Introduction (lines 42-45, 56-60) and Discussion. We also now summarise the practical contributions in a standalone paragraph in the Discussion (lines 504-522).

Reviewer 2 Report

Comments and Suggestions for Authors

First of all, thank you for the opportunity to review this work.

The manuscript contributes to understanding psychological responses to home-working practices and investigates their relationship to engagement in health and well-being-related behaviours. A sample of 491 homeworkers completed a survey that assessed ten psychological responses to homeworking, ten indices of health behaviour and seven indices of well-being, showing a complex system of potential health and well-being consequences of working from home.

This manuscript meets the journal's criteria. Although it addresses an interesting topic, I suggest improving its quality.

The introduction has been highlighted and is well-placed in the current literature. The main questions and assumptions are clearly defined. The model proposed by the authors is original and provides relevant new insights into the field of psychological responses to homework practice. 

The authors also included the existing literature on the subject, providing a theoretical basis for the study. However, based on my reading, the authors should better describe the study's novelty and how it could fill the research gaps. Therefore, I suggest improving the discussion on this aspect. A clear definition of this gap would strengthen the justification of the current research. The discussion should be more explicitly linked to recent literature, reinforcing the contribution of the current study to the broader field. In addition, a more in-depth and supported exploration of the implications of the findings for psychologists or others working in the specific field would be helpful; the authors in the conclusions talk about “public health initiatives”, and it would be good to better clarify this by explaining the implications for stakeholders and policymakers. 

Thank you, and good luck with your research.

Author Response

R2 COMMENT 1:

Based on my reading, the authors should better describe the study's novelty and how it could fill the research gaps. Therefore, I suggest improving the discussion on this aspect. A clear definition of this gap would strengthen the justification of the current research. The discussion should be more explicitly linked to recent literature, reinforcing the contribution of the current study to the broader field.

OUR RESPONSE TO R2 COMMENT 1: We now more clearly justify the study, including with reference to theory and a greater amount of (recent) empirical literature, throughout the Introduction. We have also amended the end of the Introduction to more clearly orientate the study within the current literature (lines 90-93):

“Previous research has established that home-working can potentially affect health, health behaviour and wellbeing (e.g., Keightley et al., 2022; Oakman et al., 2020; Wilms et al., 2022). This study sought to explore how home-working may have such impact, with an especial focus on workers’ psychological responses to home-working.”

We have also improved the Discussion, by relating our findings back to theory, and offering a newly-standalone paragraph setting out the practical implications of our study.

R2 COMMENT 2:

In addition, a more in-depth and supported exploration of the implications of the findings for psychologists or others working in the specific field would be helpful; the authors in the conclusions talk about “public health initiatives”, and it would be good to better clarify this by explaining the implications for stakeholders and policymakers. 

OUR RESPONSE TO R2 COMMENT 2: Our study is exploratory, and due to the methodological limitations of the study (which are now more extensively described in the Discussion), our findings should be seen as preliminary, and warrant further replication. In this context, we are keen not to overplay the implications of our findings for public health policy or practice. We have added some clarification to the paper regarding the public health implications, but only in so far as we ‘urge public health bodies to seek to raise awareness of the potential for home-working to adversely impact health and wellbeing, and thereby encourage employers to adopt more health-conducive home-working policies’ (lines 507-509). We focus the more substantial discussion of our recommendations for practice on the specific actions that employers could take (lines 509-522).

Reviewer 3 Report

Comments and Suggestions for Authors

This is generally a well structured paper with a good contribution being made through the findings. The strength of the paper is tied to the use of network analysis within the examination of the wellbeing of persons who work from home. However, there are some elements which can be further enhanced to strengthen the paper. These include:

1. A strong introduction that is rooted within a contextual framing of the study. There is little indication of where the study is located and whether there is any connection between the context and the observations being made through the data. 

2. There is no discussion or presentation of a conceptual or theoretical framework for the study. Given this, it is imperative that the authors speak to the implications of these findings for advancing work in this field and for developing such frameworks or perspectives to move this discussion forward. 

3. Incomplete description of the data collection process. While there is a detailed description of the measures and the scales, we do not know of the process for accessing these participants and of their socio-demographic backgrounds.

4. Limited testing and presentation of data-there is no comparison of the measures based on the socio-demographic background of participants. This can also help identify some patterns in the data.

5. Discussion-There is a need to speak to the implications of the work for developing theoretical and practical insights. The author can develop this further within this discussion.

Author Response

R3 COMMENT 1:

A strong introduction that is rooted within a contextual framing of the study. There is little indication of where the study is located and whether there is any connection between the context and the observations being made through the data.

OUR RESPONSE TO R3 COMMENT 1: We cannot be sure whether there was any connection between elements of the context – e.g. when and where the study was undertaken – and the findings of the study. However, we do not see this as part of the rationale for the study, so have chosen not to address it in the Introduction; instead, we highlight this as a limitation of the study in the Discussion (lines 523-541). We also now adopt a more cautious tone throughout the paper, stating that the study requires replication due to its methodological limitations, including that the data were collected when some Covid-19 restrictions were still in place (lines 523-529).

R3 COMMENT 2:

There is no discussion or presentation of a conceptual or theoretical framework for the study. Given this, it is imperative that the authors speak to the implications of these findings for advancing work in this field and for developing such frameworks or perspectives to move this discussion forward. 

OUR RESPONSE TO R3 COMMENT 2: We now clarify that our model was built inductively, drawing on our earlier qualitative research, rather than derived from theory (lines XX-XX). Nonetheless, we also now refer to relevant theory – i.e., the COM-B model, and the Job Demands-Resources Model – in the Introduction. In the Discussion, we set out implications of the findings for the Job Demands-Resources model (lines 462-474):

From the perspective of the Job Demands-Resources model (Bakker & Demerouti, 2017), which proposes that workers must possess sufficient workplace and personal resources to cope with job demands, our findings demonstrate potential interplay between work demands and coping resources. They also show how, in line with the COM-B model of behaviour (Michie et al., 2011), home-working practices can diminish opportunities for health-promoting behaviours. For example, workload pressure, a key job demand, can increase the time and psychological effort that workers spend on work-related matters in non-work time, so reducing psychological recovery from work (Hallman et al., 2021; Tejero et al., 2021). Furthermore, where managing workload involves engaging in more computer-based work, this reduces opportunities for movement and so will likely increase sitting time, and engaging in such health-risk behaviour can compromise the wellbeing needed to cope with heightened work demands (Hallam et al., 2018; ten Broeke et al., 2023; Tzaneti, 2021).”

R3 COMMENT 3:

Incomplete description of the data collection process. While there is a detailed description of the measures and the scales, we do not know of the process for accessing these participants and of their socio-demographic backgrounds.

OUR RESPONSE TO R3 COMMENT 3: We now describe the eligibility criteria for our study (which were also the same criteria as for the preliminary, item generation work) on lines 108-112. We describe how we recruited them (i.e. via Prolific.ac, an online participant recruitment platform) on lines 106-108. We summarise demographic information briefly in the commentary (lines 120-125), but note that full demographic information is provided in Supplementary Table 2.

R3 COMMENT 4:

Limited testing and presentation of data-there is no comparison of the measures based on the socio-demographic background of participants. This can also help identify some patterns in the data.

OUR RESPONSE TO R3 COMMENT 4: We had no a priori reason to expect sociodemographic differences in scores on the variables we measured, so we did not run these analyses within the Network Analysis. However, in the interests of openness, we have now run these analyses and report them as Supplementary Analysis. (While some differences were found, we have not adjusted our Network Analysis or findings because there is no a priori justification for retrofitting our inductively derived model to the findings of post-hoc analyses.)

R3 COMMENT 5:

Discussion-There is a need to speak to the implications of the work for developing theoretical and practical insights. The author can develop this further within this discussion. OUR RESPONSE TO R3 COMMENT 5: In the Discussion, we now consider the implications of our findings in light of the two theoretical models that we now cite in the Introduction (i.e., the COM-B model and Job Demands-Resources model). We also offer an expanded, stand-alone paragraph in the Discussion, setting out the practical implications (lines 504-522).

Reviewer 4 Report

Comments and Suggestions for Authors

Dear authors,

Thank you very much for the opportunity to review your article. I consider it to be a very important topic. Indeed, as you mention, remote work is becoming increasingly prevalent, and it is crucial to understand its impact on the health and well-being of workers. In this regard, I read your article with great interest.

However, my review of your article raises some concerns regarding both theoretical and methodological issues. Therefore, I take the liberty of offering some comments and suggestions that I believe could improve your work.

In theoretical terms, the paper does not propose any integrative theory or conceptual model. As a result, the relationship between variables and the study hypotheses (which are also missing) lacks any framework, so the analysis of the results becomes merely a statistical exercise without understanding why certain dimensions of Psychological Responses to Home-Working Practices are specifically related to certain indicators of Indicators of Health, Health Behaviour, and Wellbeing. In this regard, I suggest that you

explore, for example, the Conservation of Resources Theory (Hobfoll, 1989) and the Job Demands-Resources Model (Bakker & Demerouti, 2024) to build a theoretical framework for your work, as well as to establish hypotheses.

In methodological terms, I believe that this work presents some weaknesses that the authors should address in order to increase the rigor of the work they present. Regarding the preliminary study that allowed the generation of the items for Psychological Responses to Home-Working Practices, the authors should present all relevant psychometric information, namely:

1. A table with information on the loadings of the items in the different

dimensions.

2. Information on the adequacy of the data (e.g., KMO and Bartlett statistics).

3. Information on the confirmatory factor analysis supporting the proposed

multidimensional structure.

4. Cronbach’s alpha for each dimension.

Considering that this is a cross-sectional study, the authors should also present all relevant information on the strategies carried out to control the common method variance, namely:

1. Harman’s single factor test.

2. Latent factor method.

3. Measures of convergent and discriminant validity.

I hope you find these suggestions constructive and beneficial in refining your article. Your dedication to advancing the field of work health and well-being is admirable, and I believeaddressing these concerns will enhance the overall quality and impact of your work.References

Bakker, A. B., & Demerouti, E. (2024). Job demands–resources theory: Frequently asked questions. Journal of

Occupational Health Psychology, 29(3), 188–200. https://doi.org/10.1037/ocp0000376

Hobfoll, S. E. (1989). Conservation of Resources. American Psychologist, 44(3), 513–524.

https://doi.org/10.1037/0003-066x.44.3.513

Author Response

R4 COMMENT 1:

In theoretical terms, the paper does not propose any integrative theory or conceptual model. As a result, the relationship between variables and the study hypotheses (which are also missing) lacks any framework, so the analysis of the results becomes merely a statistical exercise without understanding why certain dimensions of Psychological Responses to Home-Working Practices are specifically related to certain indicators of Indicators of Health, Health Behaviour, and Wellbeing. In this regard, I suggest that you

explore, for example, the Conservation of Resources Theory (Hobfoll, 1989) and the Job Demands-Resources Model (Bakker & Demerouti, 2024) to build a theoretical framework for your work, as well as to establish hypotheses.

OUR RESPONSE TO R4 COMMENT 1: As we now explain, we built our model inductively, using a ‘bottom-up’ approach that began with our earlier qualitative study, which generated the variables that we entered into our item generation process. Nonetheless, we recognise the importance of offering links to theory, so now refer to both the COM-B model of behaviour, and the Job Demands-Resources model, in the Introduction (lines 56-71) and Discussion (lines 462-474).

R4 COMMENT 2:

Regarding the preliminary study that allowed the generation of the items for Psychological Responses to Home-Working Practices, the authors should present all relevant psychometric information, namely:

2.1. A table with information on the loadings of the items in the different

dimensions.

RESPONSE TO COMMENT 2.1: This is now provided in Supplementary Table 1.

2.2. Information on the adequacy of the data (e.g., KMO and Bartlett statistics).

RESPONSE TO COMMENT 2.2: We now report the KMO and Bartlett statistics in the main text (lines 151-152).

2.3. Information on the confirmatory factor analysis supporting the proposed multidimensional structure.

RESPONSE TO COMMENT 2.3: We did not conduct a Confirmatory Factor Analysis (CFA) to confirm our multidimensional structure. Network Analysis offers a (more exploratory) alternative to CFA’s more hypothesis-testing approach – and importantly, the two analysis forms do not both need to be conducted on the same data (e.g., Suwartono & Bintamur, 2019; https://doi.org/10.24123/aipj.v34i3.2300). That is, even if CFA and Network Analysis were undertaken on the same dataset, and the CFA findings confirmed the Network Analysis, this would not confirm that the factors identified in the dataset are replicable. Thus, the utility of conducting both CFA and Network Analysis on the same data are questionable; it is more conventional for the two methods to be used separately, to independently verify the results of each other (see e.g., Picco et al., 2020; https://doi.org/10.1080/09658211.2020.1812662).

Note that we took several steps to ensure that our Network Analysis was robust, and to mitigate for measurement error, including accuracy and stability testing, common method bias testing, and reliability analyses. We openly acknowledge the limitations of our study at several points in the paper, and argue that replication of our findings is needed. We would expect the use of Confirmatory Factor Analysis in follow-up studies that seek to replicate the findings of our research.

2.4. Cronbach’s alpha for each dimension.

RESPONSE TO COMMENT 2.4: This is now reported in Supplementary Table 1.

R4 COMMENT 3. Considering that this is a cross-sectional study, the authors should also present all relevant information on the strategies carried out to control the common method variance, namely:

COMMENT 3.1. Harman’s single factor test.

RESPONSE TO COMMENT 3.1: We now report this in the main text (lines 250-254):

Harman’s single-factor test, conducted to assess for the presence of common method variance (Podsakoff et al., 2003; Jordan & Troth, 2020), indicated that a single factor accounted for 18.61% of the variance (SS loadings = 14.52). This figure was substantially below the 50% threshold, so indicated that common method bias was not a significant concern.”

COMMENT 3.2. Latent factor method.

RESPONSE TO COMMENT 3.2: We did not conduct a Confirmatory Factor Analysis, so this is not relevant. (See our Response to Comment 2.3.)

COMMENT 3.3. Measures of convergent and discriminant validity.

RESPONSE TO COMMENT 3.3: While we recognize the importance of evaluating discriminant and convergent validity for measured constructs, we have not reported these analyses because they are not in keeping with our methodological approach. Our aim was not to use confirmatory approaches, such as discriminant and convergent validity assessments, to show predefined latent measurement structures. Rather, our analysis is exploratory, seeking to identify the optimal network of relationships between variables. Indeed, the analysis shows that clusters of variable scores emerge, suggesting that there is some convergence between, for example, variables that pertain to home-working independence (i.e., ‘pressure to work at same time as colleagues’, ‘homeworking autonomy’, ‘pressure to attend meetings’). Similarly, there is convergence among variables that pertain to the home-work transition (‘ability to switch off from work’, ‘freedom to transition between environments’, ‘ability to transform workspaces’). We do not see such convergence as problematic, but rather see it as a finding of potential interest, indicating discrete clusterings of variables underpinning how people psychological respond to home-working.

For full transparency however, and to support future validation efforts, we have included a correlation matrix of the latent home-working measures in our supplementary materials (Supplementary Table 5). This correlation matrix allows curious readers to assess potential convergence and discrimination between variables.

Round 2

Reviewer 1 Report

Comments and Suggestions for Authors

Despite having some reservations regarding some points of the article (notably in the introduction, in the discussion of results/conclusion), I consider that the article is much better and I welcome the authors' efforts to make it more suitable for the magazine.

Therefore, I believe the article is ready to be published.

Author Response

Comment: I believe the article is ready to be published.

Reply: We note that the reviewer has not requested any further edits. We thank the reviewer for their positive appraisal of our manuscript. 

Reviewer 3 Report

Comments and Suggestions for Authors

Dear Editor, 

Substantive changes have been made to this article, which address the issues that were flagged in the previous review.  However, the process for recruitment still remains unclear. Was this done through a flyer? Listerv? This is still not addressed in the methods section of the paper. It is recommended that authors address this. The recommendation therefore is for minor changes. 

Author Response

Comment: The process for recruitment still remains unclear. Was this done through a flyer? Listerv? This is still not addressed in the methods section of the paper. It is recommended that authors address this.

Response: We have now clarified recruitment processes for both the main study and the preliminary dataset, for both of which participants were recruited via Prolific.com, an online participant recruitment platform (Lines 106-115; Lines 156-160).

Reviewer 4 Report

Comments and Suggestions for Authors

.Dear Authors,

I appreciate the substantial improvements you have made to the work presented. I am grateful for your effort in incorporating many of the comments and suggestions I provided. I hope that further developments will follow to enhance the consistency and reliability of the approach used in this study.

Author Response

Comment: I appreciate the substantial improvements you have made to the work presented. I am grateful for your effort in incorporating many of the comments and suggestions I provided. I hope that further developments will follow to enhance the consistency and reliability of the approach used in this study.

Response: We note that the reviewer has not requested changes to our manuscript. We thank the reviewer for their positive appraisal of our manuscript.